# Heterologous prime-boost immunization based on a human adenovirus 5 vectored containing *Trichinella spiralis* Cystatin-like protein elicits protective mucosal immunity in mice

Nuo Xu[1,☉], Yang Wang[1,☉], Ning Xu[1], Zhenhuan Xiang[1], Dexian Wang[1], Yao Yu[1], Mingyuan Liu[1,2], Xiaolei Liu[1], Bin Tang[1]*, Xue Bai [1]*

**1** State Key Laboratory for Diagnosis and Treatment of Severe Zoonotic Infectious Diseases, Key Laboratory for Zoonosis Research of the Ministry of Education, Institute of Zoonosis, and College of Veterinary Medicine, Jilin University, Changchun, China, **2** Jiangsu Co-innovation Center for Prevention and Control of Important Animal Infectious Diseases and Zoonoses, Yangzhou, China

☉ These authors contributed equally to this work.
* tangbin1985@jlu.edu.cn (BT); namiya23@163.com (XB)

## Abstract

Trichinellosis is a globally distributed zoonotic parasitic disease. The *Trichinella* infective larvae migrate through the intestine after ingestion and settle in muscles, thus intestinal mucosal immunity plays a vital role against early infection with *Trichinella*. In this study, a recombinant adenovirus vector expressing the cysteine protease inhibitor of *Trichinella spiralis* (rAd5*Ts*CLP) was constructed and combined with the recombinant protein r*Ts*CLP in a heterologous prime-boost regimen. The regimen elicits strong, specific, and neutralizing antibodies in BALB/c mice, significantly enhancing cellular immunity through Th1 (IFN-γ, TNF-α) and Th2 (IL-13, IL-4) cytokine production in the peripheral blood, spleen, and cervical lymph nodes, driven by the activation of CD4+ and CD8+ T-cells. Notably, immunization with rAd5*Ts*CLP:r*Ts*CLP elevated mucosal secretory IgA (sIgA) levels, boosted histamine concentrations, and increased goblet cell numbers in the intestinal epithelium. Vaccinated mice showed a significant 61.17% reduction in adult worms and a 58.22% reduction in muscle larvae after the *T. spiralis* challenge. The adenovirus vector-delivered *Ts*CLP amplifies local mucosal immunity, eliciting a Th1/Th2 mixed immune response that facilitates the expulsion of *T. spiralis*. Our study provides a feasible and promising approach for *Trichinella* vaccines, further highlighting the potential of an adenovirus vector for anti-helminth vaccine development.

## Author summary

Trichinellosis, a zoonotic parasitic disease caused by *Trichinella spiralis*, poses a significant threat to global food safety and public health. Current vaccine

**Data availability statement:** All data are fully available within the manuscript, Supporting information, and Figshare (DOI: 10.6084/m9.figshare.28822889).

**Funding:** The study was supported by the National Key Research and Development Program of China, 2023YFE0107300 to XB, the National Key Research and Development Program of China, 2023YFD1802400 to BT, the National Natural Science Foundation of China, NSFC32230104, 32373032 and 82201959 to ML, the Science and technology talents and platform plan of Yunnan province, Academician and Expert Workstation, 202305AF150167 to XL. The funders had no role instudy design, data collection and analysis, decision to publish, or preparation of the manuscript.

**Competing interests:** The authors have declared that no competing interests exist.

strategies face a critical limitation in their inability to effectively induce mucosal immunity, particularly during the early intestinal invasion phase of the parasite. To address this challenge, we developed a recombinant adenovirus vaccine (rAd5*Ts*CLP) targeting the *T. spiralis* cystatin-like protein (*Ts*CLP). Though a heterologous strategy (intranasal priming with adenoviral vaccine followed by intramuscular with recombinant protein vaccine) effectively induced high levels of specific serum antibodies(IgG, IgG1, IgG2a, IgM) and neutralizing antibodies, with peak of IFN-γ, IL-13, and IL-4 in serum, spleen and cervical lymph node. Furthermore, elevated levels of secretory IgA (sIgA), histamine, and goblet cell proliferation were observed. After challenge, immunized mice showed significant protective efficacy, with adult worm and muscle larvae burdens reduced by 61.17% and 58.22%. These findings highlight the potential of adenoviral vectors to enhance local mucosal immunity and induce mixed Th1/Th2 responses, offering a promising prime-boost strategy for the development of *Trichinella* vaccines.

## Introduction

Trichinellosis is a globally prevalent parasitic disease caused by the consumption of undercooked or raw meat containing *Trichinella spiralis* (*T. spiralis*) larvae [1]. *Trichinella* infections have been identified in over 150 animal species, including humans, where it not only poses a direct threat to public health but also undermines food safety through zoonotic transmission, leading to significant socioeconomic consequences [2]. In 2014, The Food and Agriculture Organization of the United Nations (FAO) and WHO listed *T. spiralis* as number one among 24 economically and internationally traded 24 foodborne parasites [3]. Therefore, a safe and effective vaccine to prevent and control *T. spiralis* infection is essential [4].

Trichinella completes its entire life cycle within a single host through three developmental phases: adult worms (AD), newborn larvae (NBL), and muscle larvae (ML) [5]. *Trichinella spiralis* infection begins with the ingestion of meat containing infective muscle larvae (ML), which then migrate to the intestine and develop into intestinal infective larvae (IILs) [6]. After four molts, they mature into adults and produce NBL, which migrate to the muscle tissue, forming cysts, which are then consumed by new hosts [7]. Many studies have shown that the IILs invasion triggers local mucosal immune responses characterized by the recruitment and activation of lymphocytes and other immune cells, which then result in the secretion of effector molecules and mucosal IgA, mediating a strong Th2-type protective immunity that expels *Trichinella* worms [8–10]. Thus, *Trichinella* vaccine development through oral or intranasal administration can enhance efficacy by targeting the intestinal infection phase [11].

Adenovirus is a non-encapsulated spherical structured virus with a diameter of about 70–90 nm, consisting of 252 capsid particles arranged in an icosahedral arrangement and containing linear double-stranded DNA molecules inside [12,13]. Adenovirus vectors have emerged as promising vaccine platforms due to their genetic stability and ability to induce robust immunity with a single dose, driving

synchronized mucosal and systemic responses *via* oral or intranasal routes [14,15]. These vaccines are effective against infectious diseases such as Ebola and MERS and have shown exceptional protective efficacy in COVID-19 clinical trials [16,17]. Researchers are also developing adenovirus-vectored vaccines for animal diseases such as swine fever, rabies, and avian infectious bronchitis [18–20]. Adenovirus platforms also show promise against parasitic diseases, with preclinical studies highlighting strong antiparasitic effects. A recent study showed that a single Ad5-KH vaccination in BALB/c mice inhibited spleen parasite growth by 66% in visceral *leishmaniasis* [21]. The combinatorial administration of AdV-ASP2 and AdV-TS vaccines conferred complete protection against *Trypanosoma cruzi* infection in mice models, achieving 100% survival rates [22]. These findings highlight the potential of adenoviral vectors for helminth vaccine development.

Previous work in our laboratory showed that *T. spiralis* cystatin-like protein (*Ts*CLP) is one of the highly expressed immunodominant antigens, as selected by immunoscreening a cDNA library from 6-hour intestinal infective larvae [23,24].

This *Ts*CLP is expressed in all the developmental stages of *T. spiralis*, with specific localization in β-stichocytes, suggesting a role in parasite growth and reproduction. Notably, CLP contains a tyvelose structure, and it is known that anti-tyvelose monoclonal antibodies prevent *T. spiralis* larvae from invading epithelial cells, explaining its strong antigenicity [25]. An oral *Salmonella*-delivered *T. spiralis* cystatin vaccine reduced the parasite's reproductive rate by 91% *via* mucosal immunization in BALB/c mice [26]. Similarly, a yeast-expressed multi-cystatin-like domain vaccine achieved a 46.9% reduction in larval burden [27]. Collectively, these studies indicate that *Ts*CLP is a promising vaccine antigen for preventing *T. spiralis* infection.

In this study, we constructed a recombinant human adenovirus type 5 (Ad5) vector encoding *Ts*CLP (rAd5*Ts*CLP) and used it in a heterologous prime-boost immunization strategy with a recombinant *Ts*CLP protein booster (r*Ts*CLP). The multi-modal approach (intranasal prime and intramuscular boost), along with the rAd5*Ts*CLP:r*Ts*CLP regimen elicited a local mucosal secretory IgA (sIgA) and a balanced Th1/Th2 immune profile in BALB/c mice, significantly reducing adult worm burdens and providing partial protection against *T. spiralis* infection.

## Materials and methods

### Ethics statement

Female BALB/c mice (6–8 weeks old) and Wistar rats were obtained from the Norman Bethune University of Medical Science (NBUMS), China. All animal experimental procedures approved by the Animal Welfare and Research Ethics Committee of Jilin University (IACUC permit number: 20170318, Ethical approval number LSXK2019023). This study strictly adhered to the Guidelines for Ethical Review of Animal Welfare (National Standard GB/T 35892–2018, China).

### Parasite and cells

The ISS534 strain of *T. spiralis* was maintained in female Wistar rats by oral administration of 3000 larvae [28], and subsequently used for challenge infections in mice. Muscle larvae (ML) were harvested 35 days post-infection from infected mice muscle tissue *via* the pepsin-HCl digestion method [29].

HEK293A cells, preserved in our laboratory, were cultured in DMEM (Sigma-Aldrich, USA) with 10% fetal bovine serum (FBS) and 1% penicillin-streptomycin (Solarbio, China).

### Delivery siRNA into *Trichinella spiralis* muscle larvae

Chemically synthesized small interfering RNAs (siRNAs) targeting *Ts*CLP (GenBank: EU263325.1) were obtained from Sangon Biotech (Shanghai, China). Three siRNA sequences were designed: siRNA-387, siRNA-834 and siRNA-1169. A fluorescein amidite FAM-labeled control siRNA was used to assess transfection efficiency and reliability. The sequences of the siRNAs are provided in S1 Table.

*T. spiralis* muscle larvae (ML) were isolated by artificial digestion, washed three times in sterile PBS, and electroporated in a 250 µL volume containing 2500 larvae, 2 µM siRNA, and 2 µL Lipofectamine 2000. Electroporation was performed using a Bio-Rad Gene Pulser Xcell system (800 V, 200 Ω, 25 µF). Where after larvae were incubated in 500 µL RPMI 1640 at 37°C with 5% $CO_2$ for 24 hours. Mice were orally inoculated with siRNA-treated ML, and intestinal-stage adult worms (Ad3) were collected at three days post-infection (dpi).

## Preparation of recombinant adenovirus vaccine rAd5*Ts*CLP

The coding sequence (CDS) of *Ts*CLP was codon-optimized for mammalian expression without altering the encoded amino acid sequence. The optimized sequence was cloned into the pShuttle-CMV-EGFP vector (Beyotime, China), designated Ad5Neg, to create the recombinant shuttle vector pShuttle-CMV-*Ts*CLP. Homologous recombination with the pAdEasy-1 adenoviral backbone in BJ5183 cells (Beyotime, China) yielded a replication-defective human adenovirus type 5 (△E1- and △E3-) carrying the *Ts*CLP. The recombinant adenoviral plasmid was linearized with *PacI* and transfected into HEK293A cells to produce recombinant adenovirus particles, named rAd5*Ts*CLP. Then, viral stocks were harvested from cell lysates *via* three freeze-thaw cycles, clarified by centrifugation, amplified in HEK293A cells, and finally purified by cesium chloride gradient ultracentrifugation. The PCR specific primers are provided in S2 Table.

## Western blot analysis

For the siRNA knockdown experiments, transfected larvae were homogenized and centrifuged to obtain crude protein extracts. Then, infected HEK293A cells were lysed in RIPA buffer to collect protein extracts for the adenovirus infection experiments. The recombinant *Ts*CLP protein (r*Ts*CLP) was prepared in our laboratory, protein samples were resolved on 10% SDS-PAGE and transferred to methanol-preactivated PVDF membranes (Roche, Germany) at 220 mA for one hour. Membranes were blocked in 5% skim milk for one hour, and then incubated overnight at 4°C with anti-*Ts*CLP monoclonal antibodies (1:2000 dilution) or anti-GAPDH (Abcam, UK). After three Tris-Borate-Sodium Tween-20 (TBST) washes, membranes were incubated with goat anti-mouse IgG-HRP secondary antibodies (Affinity Bioreagents, USA) at room temperature for two hours, followed by five additional TBST washes. Signals were analyzed using a UVP Chemstudio system (Analytik Jena, Germany).

## Immunofluorescence assay

When a 70–90% confluence was reached in the 24-well plates, HEK293A cells were infected with either Ad5Neg or rAd5*Ts*CLP, and uninfected HEK293A cells as controls. After 48 hours, the cells were fixed in 4% paraformaldehyde (500 µL/well, 20 min), permeabilized in 0.2% Triton X-100 (500 µL/well, 5 min), and washed three times with PBS. After blocking with goat serum, the cells were incubated with anti-*Ts*CLP monoclonal antibodies (1:200 dilution) at 37°C for one hour, followed by incubation with goat anti-mice IgG (1:3000 dilution, Invitrogen, USA) at 37°C in the dark for one hour. Lastly, cells were mounted with DAPI-containing antifade medium (Beyotime, China) and viewed using a confocal laser scanning microscope (Olympus, Japan).

## Transmission electron microscopy

The Ad5Neg and rAd5*Ts*CLP viral suspensions were adsorbed onto 200 mesh copper grids, stained with 1% phosphotungstic acid for one minute. Recombinant adenovirus particles were imaged using a transmission electron microscopy (Olympus, Japan).

## Animal immunization and challenge

BALB/c mice (n = 10) were randomly divided into five groups. After the priming immunization, they were boosted at week 2 *via* either intranasal administration of rAd5*Ts*CLP ($10^8$ PFU/20 µL) or intramuscular injection of r*Ts*CLP with CpG 1018 adjuvant (60 µg/100 µL). The groups as follows: Group 1 (PBS) received an equivalent volume of PBS; Group 2 (Ad5Neg)

received $10^8$ PFU Ad5Neg (empty adenovirus vector); Group 3 (Ad5Neg:rTsCLP) was primed with $10^8$ PFU Ad5Neg, boosted with 60 µg rTsCLP; Group 4 (rAd5TsCLP:rTsCLP) was primed with $10^8$ PFU rAd5TsCLP, boosted with 60 µg rTsCLP; and Group 5 (rTsCLP:rTsCLP) received 60 µg rTsCLP for both immunizations. Mice were challenged with 250 *T. spiralis* muscle larvae at week four and euthanized at week 9 (five weeks post-challenge). Blood, diaphragm muscle, spleen, cervical lymph nodes, and small intestine samples were collected to evaluate parasite burden, humoral/cellular/mucosal immunity, and T-lymphocyte populations.

## Serology assays

Serum-specific antibodies (IgG, IgG1, IgG2a, IgM, and IgA) were quantified using indirect ELISA. Purified recombinant *Ts*CLP (5 µg/mL) was coated onto 96-well plates overnight at 4°C. Plates were blocked with 5% skim milk at 37°C for 1 h. Serum samples (10-fold diluted) were added and incubated at 37°C for 1 h. After washing, HRP-conjugated goat anti-mouse IgG/IgM (1:3000 dilution) or IgG1/IgG2a/IgA (1:10000 dilution) was incubated at 37°C for 1 h. TMB substrate was incubated in the dark at 37°C for 15 min, and the reaction stopped with 2M $H_2SO_4$. Absorbance was measured at 450 nm using a microplate reader (BioTek, USA).

For neutralization antibody (NAb) assay, serum samples were heat-inactivated at 56°C for 30 minutes and then serially diluted two-fold (1:10–1:320) in 96-well plates. Then, an equal volume of rAd5TsCLP (containing 100 $TCID_{50}$ of virus) was added to each dilution and plates were incubated at 37°C for one hour. The serum-virus mixture (100 µL) was applied to confluent HEK293A cell monolayers, with the Ad5Neg and uninfected cells as controls. After 72 hours of incubation at 37°C with 5% $CO_2$, viral infection was assessed by observing fluorescent protein expression in the cells. Finally, the neutralizing antibody titers were defined as the highest dilution achieving ≥50% reduction in fluorescent foci relative to the virus-only infected controls.

## Measurement of intestine sIgA and histamine

To assess the total sIgA, specific sIgA, and histamine levels in the intestinal fluids, intestinal flush samples were collected as described [30]. Following euthanasia, the small intestine was excised and rinsed with PBS containing 1% EDTA and 1mM PMSF, and the lavage fluid was centrifuged (4°C, 5 minutes) to collect the supernatant. The total and *Ts*CLP-specific sIgA levels were measured by indirect ELISA using plates coated with crude *T. spiralis* protein (for total IgA) and *Ts*CLP protein (for specific IgA). The intestinal lavage fluid samples served as the primary antibody, and goat anti-mouse IgA-HRP as the secondary antibody, and samples were developed with TMB substrate ($OD_{450}$ measured). Histamine levels were measured at weeks zero, three, and nine using a commercial mouse ELISA kit (Elabscience, China) following the manufacturer's instructions.

## Cytokine assays

Cytokine levels were measured in the serum, spleen, and cervical lymph nodes. Serum was collected by orbital venous plexus bleeding, and blood was incubated at 4°C overnight, centrifuged, and stored at -20°C. Splenocytes were aseptically isolated, treated with RBC lysis buffer, and cultured (5% $CO_2$, 37°C) at a concentration of $5 \times 10^6$ cells/well in 24-well plates with *Ts*CLP stimulation (5 µg/mL) for 72 hours. Cervical lymph nodes were homogenized in RPMI-1640 medium under aseptic conditions, and cells were similarly stimulated with *Ts*CLP in culture. Cell supernatants from both the splenocyte and lymph node cultures were collected after 72 hours. Cytokine levels in serum (GM-CSF, TNF-α, IL-12p70, MCP-1, IL-1β, IL-4, IL-13, IL-10, IL-17, IFN-γ, TGF-β) and in cell supernatants (IFN-γ, TNF-α, IL-4, IL-13, IL-1β, TGF-β) were measured by Luminex multiplex assay.

## Flow cytometry

Mice were anesthetized with 2% sodium pentobarbital (40–50 mg/kg), and adequate anesthesia was confirmed. Blood was collected from the orbital venous plexus, and euthanasia was performed by cervical dislocation under deep

anesthesia. Following euthanasia, mouse spleens were aseptically excised and mechanically disrupted into single-cell suspensions through a 100-µm cell strainer, and after erythrocyte lysis, cells were washed and resuspended in PBS. For immunophenotyping, splenocytes were incubated with fluorochrome-conjugated antibodies (anti-mouse CD3-APC, CD4-FITC, and CD8-PE antibodies (BioLegend, USA)) for 30 minutes at 4°C in the dark. Flow cytometry was conducted on a BD FACSCanto system (BD, USA), and data was processed using the FlowJo 10 software. The T-cell subsets (CD3$^+$CD4$^+$, CD3$^+$CD8$^+$, and CD3$^+$CD4$^+$CD8$^+$ populations) were quantified by appropriate gating on live lymphocytes.

### Small intestine and muscle histopathology

After euthanasia, the diaphragm and small intestine were isolated from the mice, dissected free of surface fascia, and rinsed with ice-cold PBS. The tissues were then fixed in 4% paraformaldehyde (PFA) for 24 hours, embedded in paraffin, and sectioned. Sections were stained with hematoxylin and eosin (H&E), as well as with an Alcian Blue Periodic acid Schiff (AB-PAS) kit (Solarbio, China), and images were captured using a bright-field microscope (Olympus, Japan). For the intestinal samples, goblet cells were counted under a multiple high-power field microscope (Olympus, Japan).

### Evaluation of worm burden

Mice were infected orally with 250 *T. spiralis* muscle larvae (ML) were euthanized at 3 days post-infection for intestinal adult worm (Ad3) and at 5 weeks post-infection for muscle larvae burden (larvae per gram tissue, LPG) determination. For adult worms, small intestines were incubated in 500 mL pre-warmed (37 °C) antibiotic saline for 3 h, allowed to settle 1 h at room temperature, then sedimented Ad3 were washed and counted under a stereomicroscope. For muscle larvae, skeletal muscle was digested in 500 mL of 1% (w/v) pepsin-1% (v/v) HCl at 37 °C with stirring for 2 h, filtered, settled, washed, and ML were counted microscopically [31,32]. Worm reduction rates were calculated compare to the PBS control group.

### Statistical analysis

Statistical analysis of data was conducted in GraphPad Prism 10 with one-way and two-way ANOVAs. Data are presented as mean ± SD, with significance levels indicated as *$P < 0.05$, **$P < 0.01$, ***$P < 0.001$, ****$P < 0.0001$.

## Results

### Bioinformatics analysis and knockdown of *Ts*CLP

Bioinformatics analysis was performed with Expasy (https://web.expasy.org/protparam/) showed that the CLP protein cDNA is 1218 bp long, encoding a 406-amino-acid protein of 45.9 kDa with an isoelectric point of 5.43. The protein contains 60 positively charged and 45 negatively charged residues. SignalP 5.0 (SignalP 5.0 - DTU Health Tech - Bioinformatic Services) predicted an 18-amino acid signal peptide without a transmembrane region. The 3D structure of *Ts*CLP was visualized using PyMOL (PyMOL | pymol.org), and the antigenicity was analyzed using DNAstar (Fig 1a).

Additionally, RNA interference (RNAi) was employed to investigate the function of *Ts*CLP in *T. spiralis* invasion. Here, efficient siRNA transfection was confirmed by strong FAM-labeled siRNA fluorescence in muscle larvae (ML) after 24 hours (Fig 1b), while qPCR results showed that *Ts*CLP mRNA levels were significantly decreased by 2 µM siRNA-1169, siRNA-387, and siRNA-834 ($P < 0.01$), with siRNA-834 achieving the greatest knockdown (Fig 1c). Correspondingly, mice infected with ML treated with siRNA-834 had significantly fewer intestinal-stage adult worms (Ad3) compared to the controls ($P < 0.01$) (Fig 1d). Western blot analysis confirmed reduced *Ts*CLP protein levels in ML and Ad3 from siRNA-treated groups, while the GAPDH levels remained unchanged (Fig 1e and 1f).

### Identification of recombinant adenovirus rAd5*Ts*CLP

Using an E1/E3-deleted Ad5 vector (Fig 2a), we created a replication-defective human type 5 adenovirus that expresses *Ts*CLP *via* integration of a codon-optimized *Ts*CLP gene. The recombinant adenovirus rAd5*Ts*CLP was generated using

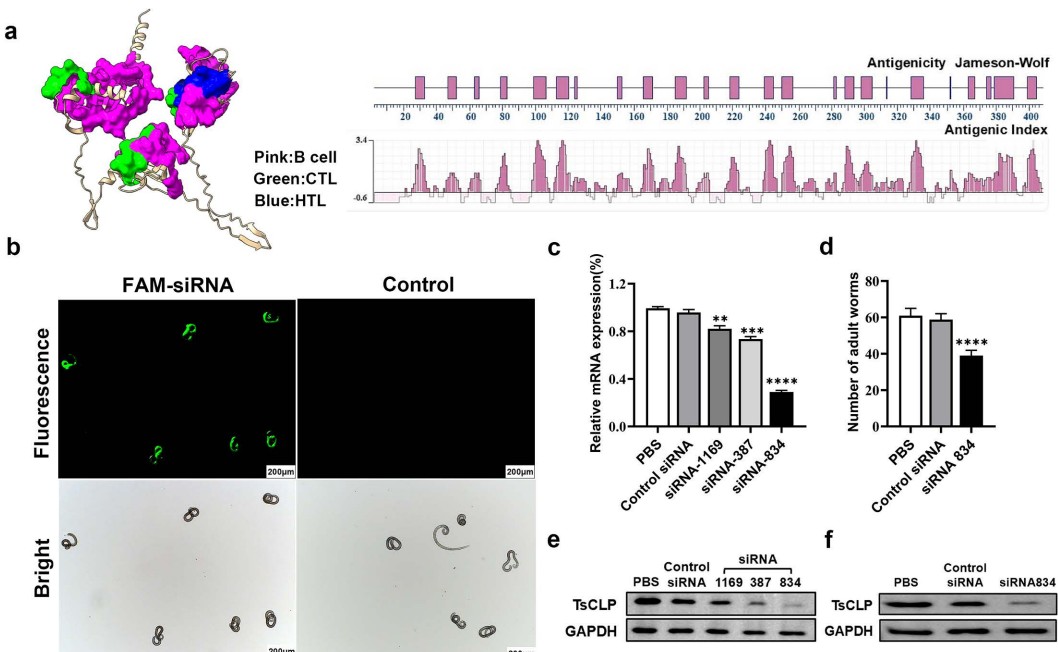

**Fig 1. Bioinformatics and gene silencing analysis of *Ts*CLP. (a)** Bioinformatics prediction of the *Ts*CLP protein 3D structure, B-cell, CTL, BTL epitopes, and antigenicity analysis. **(b)** Fluorescence microscopy detection of siRNA in *Trichinella* muscle larvae 24 hours post-electroporation. Scale bars are 200 μm. **(c)** Quantitative PCR (qPCR) analysis of *Ts*CLP mRNA levels in muscle larvae treated with three different *Ts*CLP-specific siRNAs, where siRNA-834 achieved the greatest knockdown ($p < 0.01$). **(d)** Effect of *Ts*CLP knockdown on parasite survival, shown as the number of Ad3 recovered from mice infected with siRNA-834 treated larvae. Samples showed significantly lower parasite survival than in control infections ($p < 0.01$). Western blot analysis of *Ts*CLP protein expression in muscle larvae **(e)** and adult worm **(f)** extracts following siRNA treatment. *Ts*CLP protein levels are markedly reduced in siRNA-treated samples compared to the controls (GAPDH is shown as the loading control).

the pAdEasy-1 system and rescued in HEK293A cells. Growth curve analysis at a multiplicity of infection (MOI) of 0.1 showed similar replication kinetics for rAd5*Ts*CLP and the empty vector control Ad5Neg, confirming that the inserted *Ts*CLP gene did not impair viral replication (Fig 2b). Immunofluorescence at 24 hours post-infection detected robust *Ts*CLP expression (red fluorescence) in rAd5*Ts*CLP-infected HEK293A cells, with no signal detected in Ad5Neg-infected controls (Fig 2c). The PCR and Western blot analyses also verified the presence of the 1218 bp *Ts*CLP insert and expression of the 45.9 kDa *Ts*CLP protein in rAd5*Ts*CLP-infected cells (Fig 2d). Transmission electron microscopy revealed that rAd5*Ts*CLP virions retained the typical adenoviral morphology, similar to the Ad5Neg particles, confirming successful packaging of the recombinant virus (Fig 2e).

## Serological analysis stimulated by heterologous vaccination

Previous experiments have evaluated the immunization route of rAd5*Ts*CLP (S1a Fig). The results demonstrated a similar protective efficacy and serum antibody titer for both intranasal and intramuscular routes (S1b and c Fig), notably intranasal delivery eliciting a stronger mucosal immune response (S1d Fig). Thus, we selected intranasal rAd5*Ts*CLP priming for the heterologous prime-boost strategy, which was tested in BALB/c mice (6–8 weeks old) (Fig 3a). Antibody titers serve as key correlates of vaccine-induced immune protection and results showed that in both the rAd5*Ts*CLP:r*Ts*CLP and r*Ts*CLP:r*Ts*CLP groups, IgG titers increased significantly by an order of magnitude in week one and were further elevated in week three ($P < 0.05$) (Fig 3b). The IgG1 and IgG2a titers also increased, but the IgG1 levels were consistently higher than the IgG2a. The IgG levels remained elevated through week nine, suggesting that the rAd5*Ts*CLP:r*Ts*CLP regimen induces

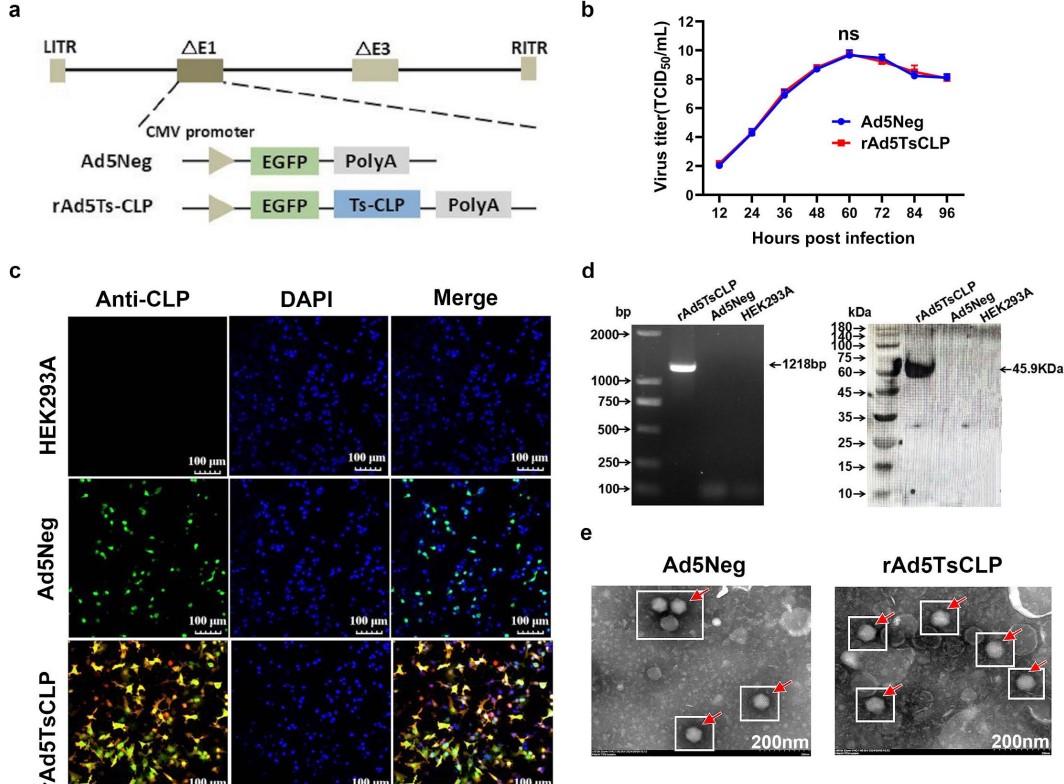

**Fig 2. Construction and characterization of the recombinant adenovirus expressing *Ts*CLP. (a)** Schematic diagram of recombinant adenovirus rAd5*Ts*CLP construction. **(b)** One-step growth curve of rAd5*Ts*CLP in HEK293A cells at a multiplicity of infection (MOI) of 0.1 compared to the empty vector, Ad5Neg. Insertion of *Ts*CLP did not affect viral replication as determined by TCID$_{50}$ assay (n = 3). **(c)** Immunofluorescence assay (IFA) showing *Ts*CLP expression (red fluorescence) in rAd5*Ts*CLP-infected HEK293A cells with no specific fluorescence observed in Ad5Neg (empty vector) infected cells. Nuclei are stained with DAPI (blue). **(d)** PCR verification of the recombinant adenovirus rAd5*Ts*CLP genome with western blot detection of *Ts*CLP expression in infected HEK293A cell lysates. **(e)** Transmission electron microscopy of purified rAd5*Ts*CLP particles showing typical adenovirus morphology. Scale bars are 200 nm.

durable immunity. Notably, IgM, which is critical for early mucosal defense, was significantly higher in the rAd5*Ts*CLP:r*Ts*-CLP group than in the Ad5Neg:r*Ts*CLP group at week one (*P* < 0.05). After the boost, IgM continued to rise, peaking at five weeks post-challenge and surpassing the levels of the other groups (*P* < 0.05), indicating sustained antigen stimulation and a robust early immune response.

Neutralizing antibody (NAb) titers peaked at week 3 in the rAd5*Ts*CLP:r*Ts*CLP group and declined by week nine, but remained significantly higher than in all the other groups, suggesting the generation of memory B cells and T-cell help for antibody maturation. Despite eliciting notable antibody responses, the homologous r*Ts*CLP:r*Ts*CLP immunization showed lower initial strength and persistence compared to the heterologous regimen. This suggests that viral vectors offer additional advantages in activating innate immunity and helper T-cells.

## Intestinal mucosal immune response elicited by rAd5*Ts*CLP:r*Ts*CLP vaccination

Mucosal immunity is typically assessed by measuring serum IgA, total sIgA, and *Ts*CLP-specific sIgA in the intestinal lavage fluid (Fig 4a). The heterologous strategy markedly enhanced mucosal immunity, with all IgA measures significantly elevated at weeks 1, 3 and 9 compared to the controls (*P* < 0.01). Histamine levels transiently spiked after immunization,

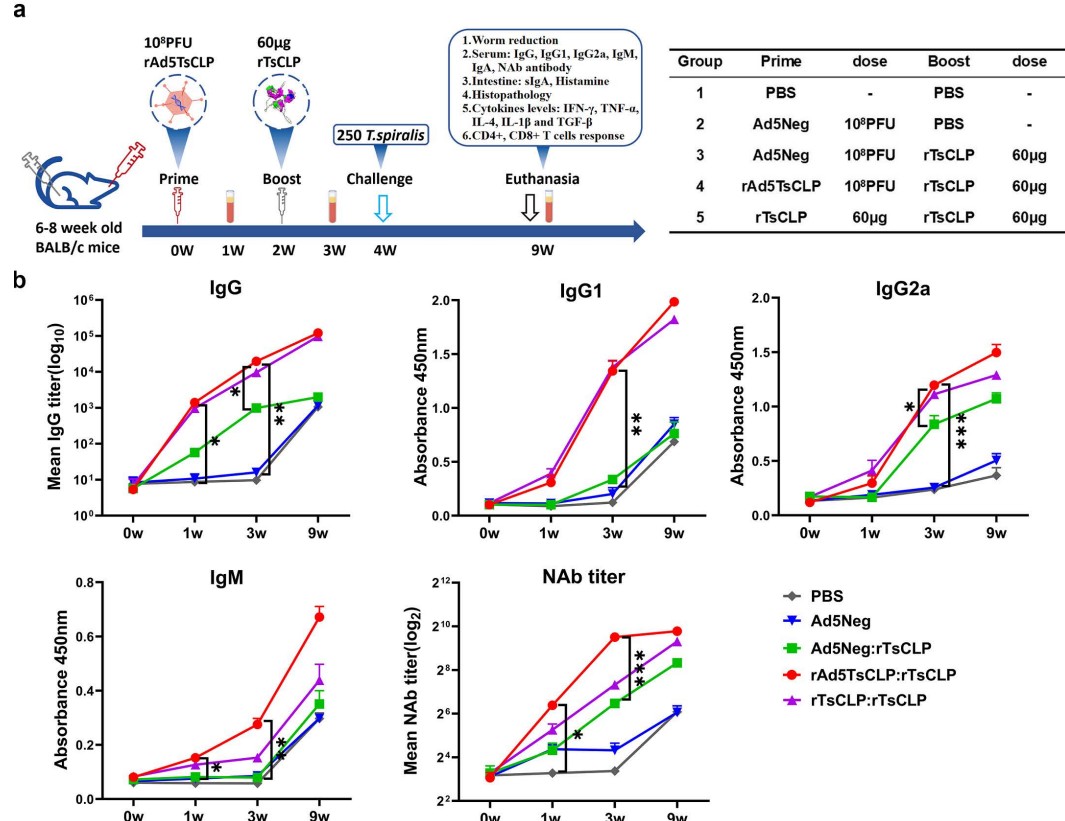

**Fig 3. Serum antibody titers induced by rAd5*Ts*CLP:r*Ts*CLP immunization.** (a) Schematic of the immunization schedule and *T. spiralis* challenge. (b) Time-course of *Ts*CLP-specific antibody responses in mice sera. Total IgG and isotype-specific IgG1 and IgG2a titers, IgM titers, and adenovirus-neutralizing antibody (NAb) titers are shown for each group. The heterologous rAd5*Ts*CLP:r*Ts*CLP group developed higher antibody responses compared to controls. Data are presented as mean ± SEM (n = 10). Statistical analysis was performed using two-way ANOVA (* $P < 0.05$; ** $P < 0.01$; *** $P < 0.001$, **** $P < 0.0001$).

correlating with acute immune cell recruitment to the mucosa, and returned to baseline by week 9 as inflammation resolved. The r*Ts*CLP:r*Ts*CLP homologous immunization induced similar mucosal responses, but with lower peak levels and shorter durability. Thus, the adenovirus prime-protein boost combination leverages the adenoviral vector's innate adjuvant effect and the protein's targeted boost to enhance B-cell responses and mucosal immune memory, leading to higher and more sustained serum IgA and sIgA levels.

Goblet cells are essential components of the intestinal epithelium, responsible for synthesizing and secreting mucins that contribute to the formation of a protective mucus barrier. In this study, AB-PAS staining was performed to simultaneously visualize goblet cells and mucin production in intestinal sections. Compared with the control groups, the rAd5*Ts*CLP:r*Ts*CLP group exhibited a substantial increase in goblet cell density, along with prominent blue-stained acidic mucin accumulation within the intestinal mucosa (Fig 4b). In contrast, other immunized groups such as r*Ts*CLP:r*Ts*CLP and Ad5Neg:r*Ts*CLP showed only moderate goblet cell expansion and relatively limited mucin secretion ($P < 0.001$). These observations suggest that the rAd5*Ts*CLP:r*Ts*CLP sequential immunization strategy not only enhances goblet cell proliferation but also preserves their secretory functionality, thereby contributing to improved mucosal barrier integrity and potentially augmenting local immune protection against *Trichinella spiralis* infection.

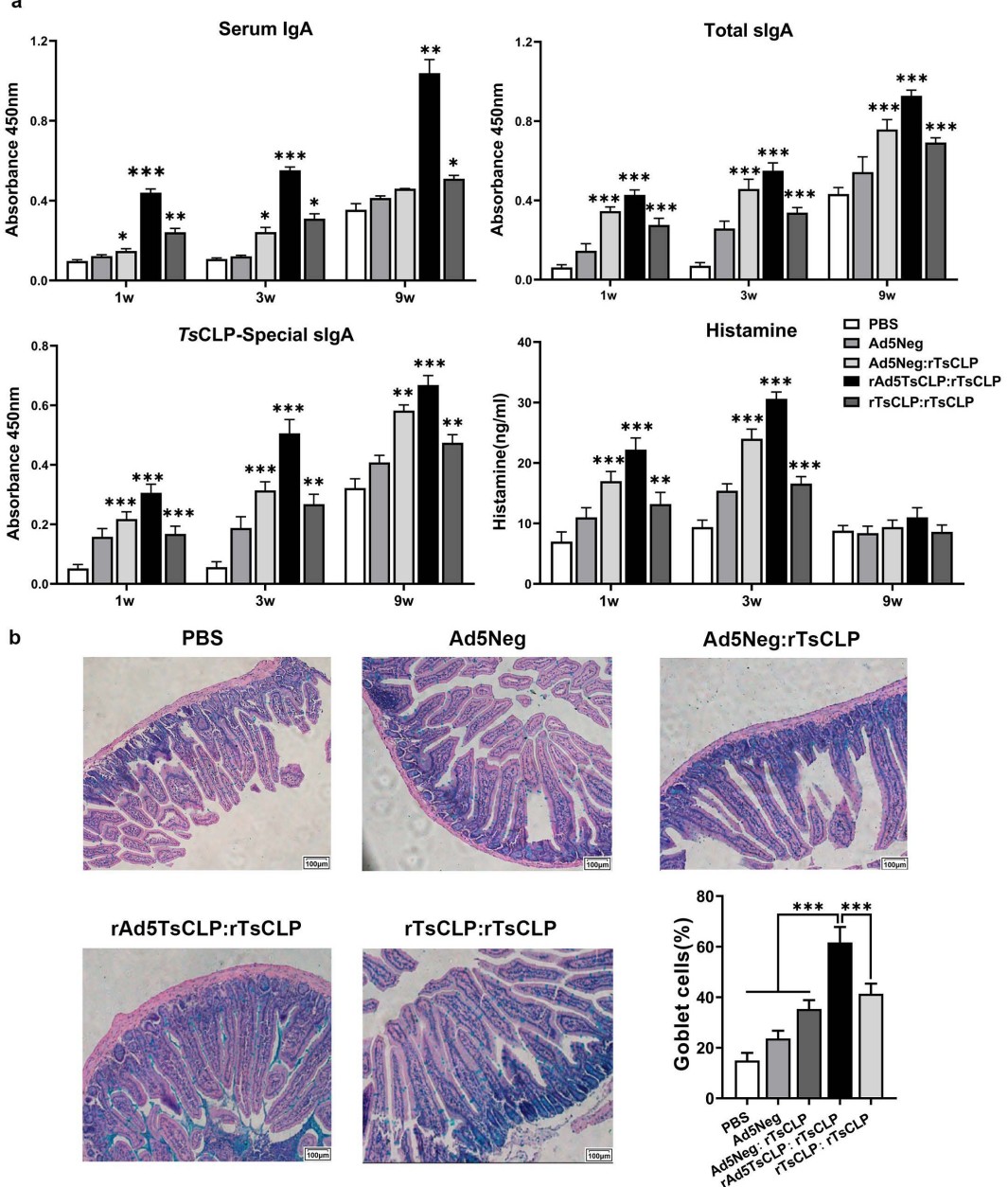

**Fig 4. Enhanced mucosal immune responses and increased goblet cell density following rAd5*Ts*CLP:r*Ts*CLP immunization. (a)** Mucosal immunity in immunized mice. Serum IgA levels, intestinal total sIgA, *Ts*CLP-specific sIgA, and intestinal histamine concentrations in intestinal lavage fluid plotted over time. The rAd5*Ts*CLP:r*Ts*CLP group shows significantly elevated IgA (systemic and mucosal) and histamine levels at multiple time points compared to other groups. Two-way ANOVA was used to compare groups (n = 10). Data are presented as mean ± SEM. P-values indicate statistical significance (*$P < 0.05$; **$P < 0.01$; ***$P < 0.001$). **(b)** Representative images of small intestinal sections from each group after challenge (AB-PAS staining). Goblet cells are more abundant in the rAd5*Ts*CLP:r*Ts*CLP-vaccinated mice compared to others. Scale bars are 100 μm. One-way ANOVA was used to compare groups (n = 10). Data are presented as mean ± SEM and P-values indicate statistical significance (*$P < 0.05$; **$P < 0.01$; ***$P < 0.001$).

## Cellular immune responses induced by rAd5*Ts*CLP:r*Ts*CLP vaccination

To characterize cellular immune responses, we profiled Th1/Th2 cytokines in the peripheral blood, spleen, and cervical lymph nodes after vaccination. Results revealed that the heterologous rAd5*Ts*CLP:r*Ts*CLP regimen induced a coordinated cytokine response across systemic and local immune compartments. In the serum, Th1 cytokines (IFN-γ, TNF-α, IL-1β, GM-CSF) and Th2 cytokines (IL-4, IL-10, IL-13) were significantly upregulated ($P < 0.001$), indicating T-cell-mediated immunity and innate immune activation, while elevated TGF-β levels further supported mucosal barrier stability ($P < 0.01$). Notably, GM-CSF increased 11-fold, suggesting enhanced activation and differentiation of myeloid-derived antigen-presenting cells. IL-12p70, a pivotal cytokine for promoting Th1 differentiation and stimulating IFN-γ production by T and NK cells, was elevated 5-fold, underscoring its central role in initiating Th1-skewed cellular immunity. In parallel, elevated MCP-1 and IL-17 indicated enhanced monocyte recruitment and a moderate involvement of the Th17 axis (Fig 5a). In the spleen, marked elevations in IFN-γ, IL-4, IL-13 ($P < 0.001$), and TNF-α ($P < 0.01$) indicated Th1/Th2 polarization and macrophage activation, while increased TGF-β levels correlated with preserved mucosal homeostasis ($P < 0.001$) (Fig 5b). In the cervical lymph nodes, robust Th1 responses (IFN-γ, TNF-α, IL-1β; $P < 0.001$) coincided with strong IL-4 and IL-13 responses ($P < 0.001$) indicative of B cell activation, and TGF-β may help limit excessive inflammation ($P < 0.001$) (Fig 5c). The heterologous prime-boost approach resulted in a broad cytokine response throughout the body and immune system, effectively integrating innate and adaptive immunity and balancing inflammatory responses. These cytokine profiles correlate with effective anti-parasitic immunity and highlight the key immune responses associated with protection.

## Antigen-specific T cell lymphocyte activity following rAd5*Ts*CLP:r*Ts*CLP vaccination

To assess the T-cell immunity in clearing *T. spiralis*, we quantified the spleen CD3⁺ T-cells in immunized mice by flow cytometry and focusing on the helper (CD3⁺CD4⁺), cytotoxic (CD3⁺CD8⁺), and double-positive (CD3⁺CD4⁺CD8⁺) subsets involved in anti-helminth responses. Compared with the PBS group, all vaccine groups significantly increased the proportions of spleen T cell subsets (Fig 6a). Notably, the rAd5*Ts*CLP:r*Ts*CLP regimen elicited the greatest expansion of CD3⁺CD4⁺, CD3⁺CD8⁺, and CD3⁺CD4⁺CD8⁺ T cells ($P < 0.001$) (Fig 6b and 6c).

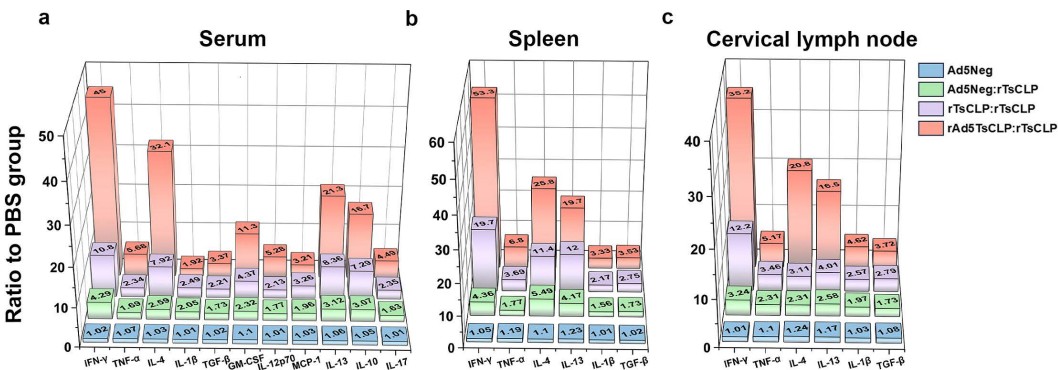

**Fig 5. rAd5*Ts*CLP:r*Ts*CLP immunization elicits a broad cytokine network. (a)** Concentrations of multiple cytokines including IFN-γ, TGF-β, IL-4, TNF-α, IL-1β, IL-13, MCP-1, IL-10, IL-17, GM-CSF, and IL-12p70 in the serum of mice five weeks post-challenge (after immunization) for each group. The heterologous regimen induces higher levels of key Th1 cytokines (e.g., IFN-γ, TNF-α), Th2 cytokines (e.g., IL-4, IL-13), and regulatory cytokines (e.g., IL-10, TGF-β) relative to the controls, indicating a mixed Th1/Th2 response. **(b)** Vitro splenocyte cultures from immunized mice, stimulated with the *Ts*CLP antigen revealed that cells from the rAd5*Ts*CLP:r*Ts*CLP group secreted significantly higher levels of IFN-γ, IL-4, and IL-13 than other groups. **(c)** Cytokine levels in the cervical lymph node cell cultures after *Ts*CLP stimulation, revealing strong local Th1 and Th2 responses in vaccinated mice. One-way ANOVA was employed to compare groups (n = 10). Data are presented as mean ± SEM. P-values indicate statistical significance (*$P < 0.05$; **$P < 0.01$; ***$P < 0.001$).

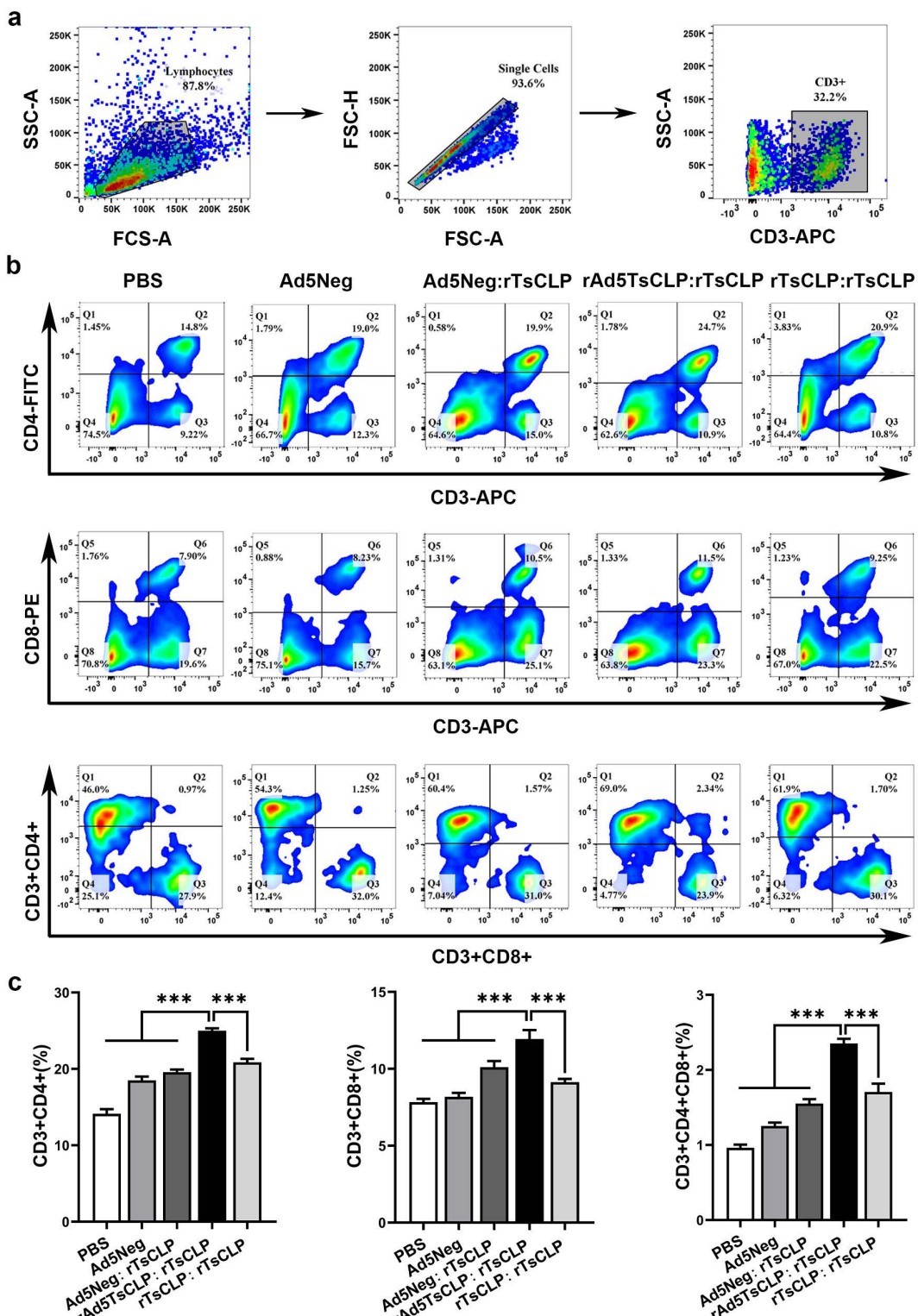

**Fig 6. Prime-boost immunization enhances T lymphocyte activation in the spleen.** (a) Flow cytometry gating strategy for identifying CD4⁺ and CD8⁺ T-cell populations among CD3⁺ splenocytes. (b) Flow cytometry plots of CD3⁺CD4⁺ T-helper cells, CD3⁺CD8⁺ cytotoxic T-cells, and CD3⁺CD4⁺CD8⁺ double-positive T-cells in the spleen of each mouse group. The rAd5*Ts*CLP:r*Ts*CLP group shows increased frequencies in all subsets compared to controls. Bar graphs represent the changes in the percentages of CD3⁺CD4⁺, CD3⁺CD8⁺, and CD3⁺CD4⁺CD8⁺ cells in each immunized group.

One-way ANOVA was performed to compare groups (n = 10). Data are presented as mean ± SEM and P-values indicate statistical significance (*P < 0.05; **P < 0.01; ***P < 0.001).

In contrast, there was no significant difference in the CD3⁺CD4⁺CD8⁺ T-cells between Ad5Neg:r*Ts*CLP and homologous r*Ts*CLP:r*Ts*CLP (*P* = 0.05) although Ad5Neg: r*Ts*CLP markedly elevated the proportion of CD3⁺CD8⁺ T-cells (*P* < 0.01). Moreover, the rAd5*Ts*CLP:r*Ts*CLP regimen elicited higher frequencies of all the T-cell subsets than the homologous r*Ts*CLP:r*Ts*CLP regimen (*P* < 0.001), underscoring the superior efficacy of the adenoviral vector-protein strategy in activating T-cell responses and enhancing multifunctional cytokine secretion. The observed increase in CD3⁺CD4⁺CD8⁺ T-cells points to the expansion of a subset likely becoming tissue-resident memory T-cells crucial for rapid mucosal immunity. However, further research is needed to elucidate their differentiation pathways.

### Vaccine-elicited protection against *Trichinella spiralis* in Mice

Vaccine-mediated protection is the gold standard for evaluating immunization strategies. BALB/c mice were orally challenged with 250 *T. spiralis* muscle larvae, and parasite burden was assessed five weeks post-infection. Compared to the PBS group, muscle larval burden were reduced by 9.88% (Ad5Neg), 21.29% (Ad5Neg:r*Ts*CLP), 58.22% (rAd5*Ts*CLP:r*Ts*CLP), and 53.39% (r*Ts*CLP:r*Ts*CLP). Adult worm (Ad3) reductions were 9.28%, 34.36%, 61.17%, and 54.3%, respectively (Fig 7a), and the heterologous rAd5*Ts*CLP:r*Ts*CLP showed slightly better efficacy than the homologous r*Ts*CLP:r*Ts*CLP. The heterologous regimen achieved a 61.17% reduction in adult worm burden, triggering a cascade of protection: by limiting intestinal adult worms establishment and reducing fertility, it significantly decreases the number of infectious muscle larvae.

Throughout the experiment, weekly weight measurements indicated no adverse clinical signs in mice, demonstrating the safety of all vaccination regimens (Fig 7b). The HE-stained diaphragm tissue sections revealed pronounced

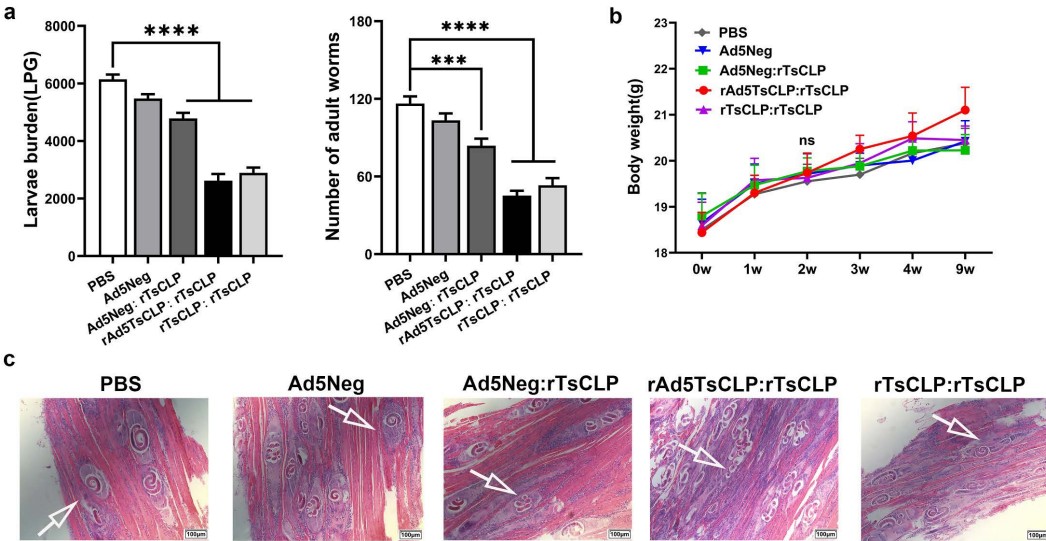

**Fig 7. Protective efficacy of vaccination regimens.** (a) Protective efficacy of immunization in mice following challenge with 250 *T. spiralis* muscle larvae. Data include muscle larva per gram (LPG) and the number of intestinal-stage adult worms (Ad3). **(b)** body weight change. (c) Hematoxylin and eosin (HE) staining of the mouse diaphragm following challenge. Scale bars are 100μm. One-way ANOVA was performed to compare groups (n = 10). Data are presented as mean ± SEM. P-values indicate statistical significance (*P < 0.05; **P < 0.01; ***P < 0.001, ****P < 0.0001).

inflammatory cell infiltration surrounding *Trichinella* cysts in the vaccinated groups, contrasting with PBS and Ad5Neg controls. Notably, the rAd5*Ts*CLP:r*Ts*CLP group exhibited the most intense inflammation, with immune cells infiltrating the cyst and directly interacting with the parasites. In the r*Ts*CLP:r*Ts*CLP group, the collagen capsule around the cysts appeared thinner, with visible damage to the encysted worms (Fig 7c).

## Discussion

Trichinellosis has become an important health concern and a food safety problem, yet despite efforts with various vaccine platforms (live attenuated, native antigen, recombinant protein, DNA, and synthetic peptide vaccines), no effective vaccine is available to prevent or control *T. spiralis* infection [33]. Currently, the challenge in developing an effective *T. spiralis* vaccine is threefold, pertaining to the antigen selection, the anti-helminth immune response and the delivery platform. Researchers have explored bacteria (e.g., *Lactobacillus* or *Salmonella*) as mucosal vectors to enhance antigen delivery and boost vaccine immunogenicity [8,34], however, they have encountered challenges including inefficient genetic modification, weak immune activation, and the need for multiple doses to sustain protection. In contrast, adenoviral vectors enable efficient transgene delivery and elicit potent immunogenicity, with a single-dose triggering coordinated mucosal and systemic immune activation. Previous work from our laboratory showed that the *T. spiralis* cystatin-like protein (*Ts*CLP), as an excretory/secretory (ES) protein, is exposed to the host's immune system early, thus activating immune pathways *via* conserved protease-binding domains. In this study, bioinformatics analysis and RNA interference confirmed CLP's antigenicity and predicted candidate epitopes for B cells, CTLs, and HTLs, further supporting its potential as a vaccine target against *T. spiralis* infection. Therefore, we constructed a recombinant adenovirus rAd5*Ts*CLP expressing *Ts*CLP and employed PCR, western blot, and immunofluorescence assays to confirm the expression of the CLP gene and protein, while transmission electron microscopy verified viral integrity. Growth kinetics assays showed that the target gene insertion was stable and did not infect the adenovirus's biological properties. These results provide a foundation for further research on the Ad5 vector's protective efficacy against *T. spiralis* infection.

In preliminary dose-response tests, a single immunization with rAd5*Ts*CLP ($10^8$ PFU), delivered either intramuscularly (IM) or intranasally (IN), elicited substantial protective efficacy in mice. Both the IM and IN routes achieved a 45.61% and 47.91% reduction in muscle larvae, and a 36.49% and 42.57% reduction in adult worms, respectively, with a significant elevation in specific and neutralizing antibody levels. Notably, the IN immunization demonstrated superior performance than compared to the IM route in stimulating serum IgA, intestinal sIgA, and histamine levels, indicating enhanced mucosal immune responses to this immunization. Thus, the data suggests that IN-administered rAd5*Ts*CLP not only strengthens mucosal barrier function but also provides effective early protection against *T. spiralis* infection. These observations align with previous studies showing that intranasal vaccination elicits robust mucosal immunity, whereas IM injection may carry a risk of vaccine-enhanced disease [35]. Thus, intranasal administration appears to be an optimal strategy for *Trichinella* vaccines, provoking dual systemic and mucosal immunity with minimal adverse effects.

For this purpose, adenoviral vectors are suitable for prime-boost strategies due to their inherent adjuvant properties, memory responses and adaptable antigen design [36]. When combined with other vaccine platforms, such as recombinant protein vaccines, they can overcome limitations like pre-existing immunity and suboptimal immunogenicity, thereby inducing stronger and more durable immune responses [37]. For example, a heterologous AdSmCB:SmCB immunization reduced *Schistosoma* burden by 71.7±7.8% in C57BL/6 mice, compared to a 42.8% reduction with homologous immunization [38]. Also, a 2023 study showed that, a "mixed modality" adenoviral prime-protein-in-adjuvant boost induced high titers of cross-reactive antibodies against *Plasmodium vivax* [39]. Therefore, the success of heterologous prime-boost regimens underscores the potential of adenoviral vectors in developing effective *T. spiralis* vaccines. In the current study, the rAd5*Ts*CLP:r*Ts*CLP prime-boost achieved a balanced efficacy-safety result, reducing muscle larval and adult worm burdens by 58.22% and 61.17%, respectively. This strategy concurrently upregulated IFN-γ and IL-4, preserving the Th1/Th2 balance and recapitulating natural antiparasitic immunity. The IL-4-driven Th2 responses

enhanced the IgE and IgG1 expression, with IgE activating mast cells and eosinophils to directly kill larvae, while IgG1 mediated antibody-dependent cellular cytotoxicity (ADCC) to help clear larvae [40]. Histopathological analysis revealed extensive inflammatory cell infiltration in the mouse diaphragm, confirming effective cell-mediated immunity. Additionally, IFN-γ-driven Th1 responses are known to increase IgG2a, which may reduce Th2 protection. However, it also activates the complimentary cleaning of free antigens, balancing the allergic risks of excessive Th2 activation [41]. Most importantly, this dynamic Th1/Th2 synergy mimics natural anti-parasitic immunity. Adenoviral vectors also accelerate IgM-NAb-IgG cascades, driving exponential increases in NAb titers and rapid complement activation, possibly through TLR9/MyD88 signaling [42]. This, in turn, optimizes the IgG1/IgG2a ratio, enhances protection and limits IgE elevation, reducing allergic risks. These findings highlight the unique ability of adenoviral vectors to improve antigen presentation and immune memory, bridging antibody-dependent and cell-mediated defenses, and advancing helminth vaccine strategies.

The intestinal infective larvae (IILs) are the first invasive stage in the life cycle of *T. spiralis*, inducing the mucosal immune protective response [43]. In this study, we also evaluated the synergistic action of sIgA and histamine and the results indicate that rAd5*Ts*CLP:r*Ts*CLP immunization significantly enhanced serum IgA, intestinal sIgA, and histamine levels, with efficacy exceeding that of recombinant protein immunization alone. The sIgA specifically binds to the larval surface antigens *via* the Fab region, preventing attachment to intestinal epithelial cells and maintaining intestinal homeostasis [44]. Together with histamine-initiated MUC2 mucin secretion by goblet cells to form a physical barrier and to alter the intestinal microenvironment, it limits larval colonization [45]. We also found that adenovirus-prime/protein-boost immunization enhanced goblet cell activity, and by simultaneously activating both the sIgA and histamine pathways, the adenoviral vector drives a "dual-engine" mucosal immunity mechanism, enhancing the overall efficacy of the immune response.

Furthermore, T cell-mediated immunity plays a pivotal role in combating parasitic infections [46]. The heterologous rAd5*Ts*CLP:r*Ts*CLP vaccine strategy synergistically enhances protection by dynamically balancing the Th1/Th2 responses and amplifying T cell activation. More specifically, the adenoviral vector potentiates MHC-I/II antigen presentation and co-stimulatory signals (e.g., CD80/86), driving a robust expansion of CD3$^+$CD4$^+$/CD8$^+$T-cells while coordinating the Th1 (IFN-γ, TNF-α) and Th2 (IL-4, IL-13) cytokine networks [47,48]. Compared to the DNA prime-protein boost regimens [49], adenovirus-mediated innate immune activation and sustained antigen exposure elicit superior T-cell responses. Additionally, elevated IFN-γ and IL-4 levels in splenic and cervical lymph node compartments synergize with cytotoxic T-cell activity and secretory IgA production to accelerate larval clearance post-challenge, while TGF-β and IL-10 maintain immunoregulatory homeostasis. Although the *Ts*CLP protein alone may suppress lysosomal proteases and promote Th2/Treg polarization in APCs, adenovirus co-delivery transforms this property into an immunogenic advantage [50]. Combining the *Ts*CLP protein with the adenovirus synergistically enhances immune regulation, boosting both humoral and cellular anti-parasite responses. The adenovirus triggers TH1 responses (IFN-γ, TNF-α), which complement *Ts*CLP-driven antibody production (IgG1, IgE). This synergy is mediated by cytokines, where: GM-CSF promotes myeloid differentiation and antigen presentation, while TNF-α and IL-1β amplify inflammatory signals to activate macrophages and neutrophils, and IL-12p70 further increases IFN-γ secretion to strengthen cytotoxic T-cell function [51]. Systemically, MCP-1 recruits monocytes and dendritic cells to infection sites, while IL-4/IL-13 drives B-cell class switching, and IL-10/TGF-β modulate inflammation to prevent tissue damage [52,53]. Moreover, intranasal immunization activates local T-cells *via* CD103$^+$dendritic cell OX40L signaling, leading to systemic diffusion of IFN-γ and IL-4, while goblet cell proliferation and sIgA secretion fortifies mucosal barriers to limit larval colonization [15,54]. Overall, this approach mimics the natural immune balance by integrating the adenovirus-mediated antigen presentation with *Ts*CLP-induced antibody responses to achieve a synergistic, multidimensional antiparasitic effect.

These adenoviral vectors act as natural adjuvants, activating pattern-recognition receptor pathways (e.g., TLRs, RLRs) to provoke broad immune responses [55]. Toll-like receptor (TLR) agonist adjuvants have demonstrated significant

advantages when used with adenoviral vectors [56]. Therefore, we selected the TLR9 agonist CpG 1018 as our adjuvant to enhance the efficacy of the rTsCLP vaccine by boosting cellular immune responses. CpG 1018 induces Th1-biased systemic and mucosal immunity through the MyD88–IRF7/NF-κB signaling pathways [57]. It has demonstrated clinical efficacy and safety as the active component of the licensed HEPLISAV-B hepatitis B vaccine, which also offers antigen dose reduction advantages [58]. Furthermore, CpG 1018 exhibits superior immunostimulatory activity compared to traditional adjuvants in both systemic and mucosal immune compartments. However, the formulation stability and the systemic reactogenicity need further optimization and future vaccine development should prioritize immunodominant antigen combinations with adjuvants that synergistically modulate Th1/Th2 balance while maintaining safety profiles. Notably, current vaccine research predominantly relies on murine models, lacking validation in natural hosts like swine, and given that pigs are the principal reservoir for *Trichinella spiralis* transmission, prioritizing porcine models in preclinical development is essential for effective vaccine assessment.

In conclusion, our study suggest that *Ts*CLP is a promising candidate antigen against *Trichinella* early infection, with intranasal immunization with rAd5*Ts*CLP eliciting a stronger mucosal and systemic immune responses compared to intramuscular injection. Additionally, in mice models, the rAd5*Ts*CLP vaccine induces superior cellular immune responses and provides a greater parasite reduction compared to previous plasmid-based formulations. The prime-boost immunization strategy enhances protective immunity over homologous r*Ts*CLP:r*Ts*CLP immunization. These results strongly support the continued development of this candidate vaccine by offering new strategies for *Trichinella* vaccine research and providing insights relevant to broader helminth vaccine development.

## Supporting information

**S1 Fig. IN induces a stronger mucosal immune response than IM inoculation in serum and intestine.** (a). A scheme of experiments. (b) Protective analyses of rAd5TsCLP immunization in mice challenged with 250 *Trichinella spiralis* muscle larvae by measuring larval burden (LPG), adult worm counts (Ad3), and changes in body weight. (c) Serum levels of IgG, IgG1, IgG2a, IgA, IgM, and NAb antibodies in mice. (d) IgA antibody level in serum, sIgA antibody and histamine level in intestinal lavage fluid of mice. P values were analyzed by one-way ANOVA for statistical differences (n = 10) (* $P < 0.05$; ** $P < 0.01$; *** $P < 0.001$, **** $P < 0.0001$).
(TIF)

**S2 Fig. The antibody levels and protective efficacy in the adjuvant-only control group.** (a) Serum levels of IgG1, IgG2a, IgA and IgM antibodies in mice. (b) Serum levels of total IgG and neutralizing antibody (NAb) titers in mice.(c) Protective efficacy of the adjuvant-only group compared to the PBS control group, evaluated by worm reduction rate. P values were analyzed by two-way ANOVA for statistical differences (n = 10) (* $P < 0.05$; ** $P < 0.01$).
(TIF)

**S3 Fig. The antibody levels and protective efficacy in the homologous immunity group.** (a) Serum levels of IgG1, IgG2a, IgA and IgM antibodies in mice. (b) Serum levels of total IgG and neutralizing antibody (NAb) titers in mice.(c) Protective efficacy of the homologous immunization group compared to the PBS control group, evaluated by worm reduction rate. P values were analyzed by two-way ANOVA for statistical differences (n = 10) (* $P < 0.05$; ** $P < 0.01$; *** $P < 0.001$, **** $P < 0.0001$).
(TIF)

**S1 Table. siRNA sequence information of *Ts*CLP.**
(XLSX)

**S2 Table. PCR specific primers.**
(XLSX)

# Acknowledgments

We would like to thank Xiaoxiao Ma for his help in the technical assistance, Xinrui Wang for her help in paraffin sectioning and staining experiments, Li Yang for her help in using the laser confocal instrument and Yuanyuan Zhang for her help in using transmission electron microscopy. Our thanks are also extended to express our gratitude to all the people who made this work.

# Author contributions

**Conceptualization:** Mingyuan Liu, Xiaolei Liu, Xue Bai.

**Data curation:** Nuo Xu, Yang Wang, Ning Xu, Bin Tang.

**Investigation:** Nuo Xu, Zhenhuan Xiang, Xue Bai.

**Methodology:** Nuo Xu, Ning Xu, Xue Bai.

**Project administration:** Nuo Xu, Zhenhuan Xiang, Dexian Wang, Yao Yu.

**Resources:** Mingyuan Liu, Xiaolei Liu, Xue Bai.

**Supervision:** Yang Wang, Ning Xu, Bin Tang.

**Writing – original draft:** Nuo Xu, Bin Tang, Xue Bai.

**Writing – review & editing:** Nuo Xu, Xiaolei Liu, Xue Bai.

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
