## [Decision Letter · Decision Letter 0]

Response to Reviewers
Revised Manuscript with Track Changes
Manuscript

Shaden Kamhawi

co-Editor-in-Chief

Paul Brindley

co-Editor-in-Chief

**Journal Requirements:**

1) Please upload all main figures as separate Figure files in .tif or .eps format. For more information about how to convert and format your figure files please see our guidelines: 

2) We have noticed that you have uploaded Supporting Information files, but you have not included a list of legends. Please add a full list of legends for your Supporting Information files after the references list.

3) Some material included in your submission may be copyrighted. According to PLOSu2019s copyright policy, authors who use figures or other material (e.g., graphics, clipart, maps) from another author or copyright holder must demonstrate or obtain permission to publish this material under the Creative Commons Attribution 4.0 International (CC BY 4.0) License used by PLOS journals. Please closely review the details of PLOSu2019s copyright requirements here: PLOS Licenses and Copyright. If you need to request permissions from a copyright holder, you may use PLOS's Copyright Content Permission form.

Potential Copyright Issues:

- Figures 3 and S1. Please confirm whether you drew the images / clip-art within the figure panels by hand. If you did not draw the images, please provide (a) a link to the source of the images or icons and their license / terms of use; or (b) written permission from the copyright holder to publish the images or icons under our CC BY 4.0 license. Alternatively, you may replace the images with open source alternatives. See these open source resources you may use to replace images / clip-art:

4) In the online submission form, you indicated that "All data are available from the corresponding author upon request.". All PLOS journals now require all data underlying the findings described in their manuscript to be freely available to other researchers, either

- In a public repository

- Within the manuscript itself

- Uploaded as supplementary information.

5) Please ensure that the funders and grant numbers match between the Financial Disclosure field and the Funding Information tab in your submission form. Note that the funders must be provided in the same order in both places as well.

**Reviewers' comments:**

**Key Review Criteria Required for Acceptance?**

**Methods:**

-Are the objectives of the study clearly articulated with a clear testable hypothesis stated?

-Is the study design appropriate to address the stated objectives?

-Is the population clearly described and appropriate for the hypothesis being tested?

-Is the sample size sufficient to ensure adequate power to address the hypothesis being tested?

-Were correct statistical analysis used to support conclusions?

-Are there concerns about ethical or regulatory requirements being met?

Reviewer #1: Yes

Reviewer #2: "Minor Revision"

The methodology applied in this study is comprehensive and well-structured, incorporating a variety of techniques for the validation of results. However, I would require clarification on certain aspects, which I outline below:

Line 119. This paragraph is somewhat unclear, as it states that parasite maintenance is carried out in Wistar rats, while larval collection is performed in mice. Could you clarify the methodology?

Line 179. Was the inclusion of an adjuvant-only control group considered in the experimental design?

Line 187. As I understand it, the booster immunizations were consistently administered intramuscularly with rTsCLP; was revaccination with rAd5TsCLP via the intranasal route considered at any point?

Line 234. It should be noted that the mice were previously anesthetized before blood collection from the orbital venous plexus."

Line 249. Was the goblet cell count performed using Hematoxylin and Eosin staining, or was another more specific stain, such as Alcian blue, used?

Line 254. It would be appropriate to specify the methodology used for the recovery of adult worms from the intestine, as well as the procedure for obtaining muscle larvae.

**Results:**

-Does the analysis presented match the analysis plan?

-Are the results clearly and completely presented?

-Are the figures (Tables, Images) of sufficient quality for clarity?

Reviewer #1: Yes

Reviewer #2: "The results align with the objectives of the study and are appropriately presented. However, the histological images, at least in the version of the manuscript reviewed, lack sufficient quality. Additionally, I would like to offer a couple of recommendations.

Line 377. Some studies suggest that Trichinella spiralis infection naturally induces an increase in goblet cell numbers; however, due to endoplasmic reticulum stress, these cells may be unable to adequately produce mucins, thereby compromising their protective role. In your study, I would recommend assessing mucin expression, as an increase in goblet cell numbers alone is not necessarily indicative of enhanced protection..

Line 410. Why were cervical lymph nodes analyzed instead of mesenteric lymph nodes?

**Conclusions:**

-Are the conclusions supported by the data presented?

-Are the limitations of analysis clearly described?

-Do the authors discuss how these data can be helpful to advance our understanding of the topic under study?

-Is public health relevance addressed?

Reviewer #1: Yes

Reviewer #2: (No Response)

**Editorial and Data Presentation Modifications?**

Reviewer #1: (No Response)

Reviewer #2: Minor revision is recommended according to the suggestions provided in the previous sections.

**Summary and General Comments:**

Reviewer #1: Reviewer comments on PNTD-D-25-00587

In this manuscript titled “Heterologous prime-boost immunization based on a human adenovirus 5 vectored containing Trichinella spiralis Cystatin-like protein elicits protective mucosal immunity in mice”(PNTD-D-25-00587), the authors dressed an important problem. The objectives of this study are clear and the experimental design is appropriate. Overall, I recommend that this manuscript could be accepted for publication after some minor revisions. They are as follows:

1. General comment: Ensure consistent use of italics for species or genus names (e.g., Trichinella spiralis,); Include in your description if you talk about larvae or adult worms (in the discussion thread as well). And the Latin names of the genus in the references, Salmonella typhimurium-> Salmonella typhimurium in italics.

2. Line 55: put a space before the square brackets [], such as larvae[1]-> larvae [1].

3. Line76. Provide a brief description of the basic structural features of adenovirus (e.g., capsid organization, genome structure) to contextualize the vector design for readers unfamiliar with adenoviral systems.

4. Line92-94. Previous published studies should be cited to support that cystatin-like proteins from Trichinella spiralis are localized in β-stichocytes and expressed in various developmental stages, including the early invasion and muscle larval stages (Tang et al., 2015; Guiliano et al., 2009).

a) Tang B, Liu M, Wang L, Yu S, Shi H, Boireau P, Cozma V, Wu X, Liu X. Characterisation of a high-frequency gene encoding a strongly antigenic cystatin-like protein from Trichinella spiralis at its early invasion stage. Parasit Vectors. 2015; 8:78. doi: 10.1186/s13071-015-0689-5.

b) Guiliano DB, Oksov Y, Lustigman S, Gounaris K, Selkirk ME. Characterisation of novel protein families secreted by muscle stage larvae of Trichinella spiralis. Int J Parasitol. 2009;39(5):515-24. doi: 10.1016/j.ijpara.2008.09.012. Epub 2008 Oct 21.

5. Line182. Please verify the selection of adjuvant CpG1018 over other adjuvants (e.g., its mechanism of action as a TLR9 agonist and relevance to mucosal immunity).

6. The authors should explicitly specify whether the control group received the same 250 muscle larvae as the infection group to ensure consistency in experimental design and rule out volume-dependent effects.

7. Line398. While cytokine profiling is comprehensive, the roles of IL-12p70, MCP-1, and IL-17 in the observed immune response should be explicitly presented in the results section.

8. Figure 6c. Correct the x-axis label "AdTsCLP:rTsCLP" as "rAd5TsCLP:rTsCLP" to align with the nomenclature used in the Methods section.

9. Suggest the authors to clarify whether the term 'Special sIgA' in Figure 4 refers to 'TsCLP-specific sIgA' and revise the figure accordingly for consistency with the legend, and to clearly state the statistical methods used in Figure 3, including the type of test and significance thresholds in the figure legend.

Reviewer #2: (No Response)

PLOS authors have the option to publish the peer review history of their article (what does this mean? ). If published, this will include your full peer review and any attached files.

**Do you want your identity to be public for this peer review?** For information about this choice, including consent withdrawal, please see our Privacy Policy .

Reviewer #1: No

Reviewer #2: No

**Figure resubmission:****Reproducibility:** To enhance the reproducibility of your results, we recommend that authors of applicable studies deposit laboratory protocols in protocols.io, where a protocol can be assigned its own identifier (DOI) such that it can be cited independently in the future. Additionally, PLOS ONE offers an option to publish peer-reviewed clinical study protocols. Read more information on sharing protocols at https://plos.org/protocols?utm_medium=editorial-email&utm_source=authorletters&utm_campaign=protocols

---

## [Decision Letter · Decision Letter 1]

Dear Dr Bai,

We are pleased to inform you that your manuscript 'Heterologous prime-boost immunization based on a human adenovirus 5 vectored containing Trichinella spiralis Cystatin-like protein elicits protective mucosal immunity in mice' has been provisionally accepted for publication in PLOS Neglected Tropical Diseases.

Best regards,

Javier Sotillo

Academic Editor

Jong-Yil Chai

Section Editor

Shaden Kamhawi

co-Editor-in-Chief

Paul Brindley

co-Editor-in-Chief

Reviewer's Responses to Questions

**Key Review Criteria Required for Acceptance?**

**Methods**

-Are the objectives of the study clearly articulated with a clear testable hypothesis stated?

-Is the study design appropriate to address the stated objectives?

-Is the population clearly described and appropriate for the hypothesis being tested?

-Is the sample size sufficient to ensure adequate power to address the hypothesis being tested?

-Were correct statistical analysis used to support conclusions?

-Are there concerns about ethical or regulatory requirements being met?

Reviewer #1: (No Response)

Reviewer #2: (No Response)

**Results**

-Does the analysis presented match the analysis plan?

-Are the results clearly and completely presented?

-Are the figures (Tables, Images) of sufficient quality for clarity?

Reviewer #1: (No Response)

Reviewer #2: (No Response)

**Conclusions**

-Are the conclusions supported by the data presented?

-Are the limitations of analysis clearly described?

-Do the authors discuss how these data can be helpful to advance our understanding of the topic under study?

-Is public health relevance addressed?

Reviewer #1: (No Response)

Reviewer #2: (No Response)

**Editorial and Data Presentation Modifications?**

Reviewer #1: (No Response)

Reviewer #2: (No Response)

**Summary and General Comments**

Reviewer #1: The revised manuscript (PNTD-D-25-00587R1) has been obviously improved. All questions and errors have been highlighted and corrected. I recommend that this manuscript can be accepted for publication in the present form.

Reviewer #2: (No Response)

PLOS authors have the option to publish the peer review history of their article (what does this mean? ). If published, this will include your full peer review and any attached files.

**Do you want your identity to be public for this peer review?** For information about this choice, including consent withdrawal, please see our Privacy Policy .

Reviewer #1: No

Reviewer #2: **Yes: ** Juan José García-Rodríguez

---

## [Editor Report · Acceptance letter]

Dear Dr Bai,

We are delighted to inform you that your manuscript, "Heterologous prime-boost immunization based on a human adenovirus 5 vectored containing Trichinella spiralis Cystatin-like protein elicits protective mucosal immunity in mice," has been formally accepted for publication in PLOS Neglected Tropical Diseases.

Best regards,

Shaden Kamhawi

co-Editor-in-Chief

Paul Brindley

co-Editor-in-Chief
